

# Changes in the force-time curve during a repeat power ability assessment using loaded countermovement jumps

Alex O. Natera[1,2], Steven Hughes[1], Dale W. Chapman[3], Neil D. Chapman[2] and Justin W.L. Keogh[2,4,5]

[1] Sport Science, New South Wales Institute of Sport, Sydney Olympic Park, New South Wales, Australia
[2] Faculty of Health Sciences and Medicine, Bond University, Gold Coast, Queensland, Australia
[3] Curtin School of Allied Health, Curtin University, Perth, Western Australia, Australia
[4] Sports Performance Research Centre New Zealand, Auckland University of Technology, Auckland, New Zealand
[5] Kasturba Medical College, Manipal Academy of Higher Education, Manipal, Karnataka, India

Corresponding author
Alex O. Natera,
alex.natera@hotmail.com

## ABSTRACT

**Background**. Repeat power ability (RPA) assessments traditionally use discrete variables, such as peak power output, to quantify the change in performance across a series of jumps. Rather than using a discrete variable, the analysis of the entire force-time curve may provide additional insight into RPA performance. The aims of this study were to (1) analyse changes in the force-time curve recorded during an RPA assessment using statistical parametric mapping (SPM) and (2) compare the differences in the force-time curve between participants with low and high RPA scores, as quantified by traditional analysis.

**Materials and Methods**. Eleven well-trained field hockey players performed an RPA assessment consisting of 20 loaded countermovement jumps with a 30% one repetition maximum half squat load (LCMJ20). Mean force-time series data was normalized to 100% of the movement duration and analysed using SPM. Peak power output for each jump was also derived from the force-time data and a percent decrement score calculated for jumps 2 to 19 ($RPA_{\%dec}$). An SPM one-way ANOVA with significance accepted at $\alpha = 0.05$, was used to identify the change in the force-time curve over three distinct series of jumps across the LCMJ20 (series 1 = jumps 2–5, series 2 = jumps 9–12 and series 3 = jumps 16–19). A secondary analysis, using an independent $T$-test with significance accepted at $p < 0.001$, was also used to identify differences in the force-time curve between participants with low and high $RPA_{\%dec}$.

**Results**. Propulsive forces were significantly lower ($p < 0.001$) between 74–98% of the movement compared to 0–73% for changes recorded during the LCMJ20. *Post hoc* analysis identified the greatest differences to occur between jump series 1 and jump series 2 ($p < 0.001$) at 70–98% of the movement and between jump series 1 and jump series 3 ($p < 0.001$) at 86–99% of the movement. No significant differences were found between jump series 2 and jump series 3. Significant differences ($p < 0.001$) in both the braking phase at 44–48% of the jump and the propulsive phase at 74–94% of the jump were identified when participants were classified based on low or high $RPA_{\%dec}$ scores (with low scores representing an enhanced ability to maintain peak power output than high scores).
**Conclusion**. A reduction in force during the late propulsive phase is evident as the LCMJ20 progresses. SPM analysis provides refined insight into where changes in the force-time curve occur during performance of the LCMJ20. Participants with the lower RPA$_{\%dec}$ scores displayed both larger braking and propulsive forces across the LCMJ20 assessment.

## INTRODUCTION

The repeat power ability (RPA) assessment, consisting of 20 repetitions of loaded countermovement jumps (LCMJ20) has been used to determine an athlete's ability to maintain peak power output (*Natera et al., 2023*). The LCMJ20 has been shown to associate with repeated high intensity effort (RHIE) performance and may be considered an underlying physical quality that can determine RHIE performance (*Natera et al., 2024*). In the RPA assessment, peak power output is extrapolated from force-time data, for each jump of the LCMJ20 and a percent decrement score is calculated to provide an RPA score (RPA$_{\%dec}$). This RPA$_{\%dec}$ is a percentage score that can reliably inform practitioners as to an athlete's ability to maintain maximal peak power output (*Natera et al., 2023*). Mean RPA$_{\%dec}$ scores for male hockey players have been found to be $26.81 \pm 3.8\%$ (*Natera et al., 2023*). Through graphical visual analysis, it appears the range of RPA$_{\%dec}$ scores is between ~20–34% (*Natera et al., 2023*). Lower RPA$_{\%dec}$ score, closer to 20%, indicates a greater ability to maintain maximal power output, while a higher RPA$_{\%dec}$ score, closer to 34%, indicates a reduced ability to maintain maximal power output.

Discrete variables like peak power output and peak velocity are commonly used to describe RPA performance and more generally vertical jump performance (*Chavda et al., 2018*; *Hori et al., 2009*; *Natera, Cardinale & Keogh, 2020*; *Warr et al., 2020*). However, discrete variables are derived from instances on the force-time curve which may not represent how the entire movement is achieved (*Preatoni et al., 2013*). Rather than relying on one instance of the force-time curve, examining the whole force-time curve may provide greater insight into an athlete's ability to maintain maximal peak power output in the LCMJ20.

It is also important to consider that as fatigue develops during an assessment like the LCMJ20, a change in jump strategy might occur as the athlete attempts to maintain power output (*Jidovtseff et al., 2014*). Although external feedback was used in an attempt to control countermovement depth and the timing of each jump during the LCMJ20, it is still possible that small but meaningful changes in jump movement strategy may occur and these changes may influence the maintenance of power output. For example, despite a metronome being used to control jump timing there may still be a change in the duration of either the braking or the propulsive phase of the jump and such changes may alter

power output (*Gathercole et al., 2015*). Examining the entirety of the force-time curve (*i.e.,* the unweighting, braking and propulsive phases of each jump), may provide a better understanding of potential changes in jump strategy that might in turn influence RPA performance.

Despite the potential to miss valuable information across the entire force-time curve, analysing one-dimensional force time data with traditional zero-dimensional methods may also increase the potential for bias (*Pataky, 2012*; *Pataky, Vanrenterghem & Robinson, 2016*). One dimensional analysis, using Random Field Theory, is said to reduce bias and Type I error by avoiding the use of separate inferential tests at each time point along the force-time curve (*Pataky, 2012*; *Pataky, Vanrenterghem & Robinson, 2016*). The use of discrete variables in zero-dimensional analysis can be considered a reductionist approach where the removal of data has the potential to introduce bias into the analysis process (*Hughes et al., 2022*). Zero-dimensional analysis may also increase the risk of Type I error and therefore the incorrect rejection of a null hypothesis (*Pataky, 2012*). Several studies have used statistical parametric mapping (SPM) as a one-dimensional analysis of the force-time curve to give greater insight and a more accurate depiction of changes in both jump and resistance exercise performance (*Hughes et al., 2022*; *Meechan et al., 2022*; *Thomas, Jones & Dos'santos, 2022*).

SPM analysis has been used to assess fatigue induced changes in vertical jump performance and to identify both force-time curve differences in weightlifting derivatives and differences in various exercises as a result of external load (*Hughes et al., 2022*; *Kipp, Comfort & Suchomel, 2021*; *Meechan et al., 2022*). When comparing SPM analysis with zero-dimensional analysis, only SPM analysis of force data was able to detect differences as a consequence of fatigue (*Hughes et al., 2022*). Thus, one dimensional analysis might be a more appropriate analysis of LCMJ20, where compounding fatigue across the assessment is likely to affect both rapid force generation and stretch shortening cycle activity (*Cormack, Newton & McGuigan, 2008*; *Hughes et al., 2022*). As SPM analysis appears to be a robust analysis method in identifying differences in movement strategy, it may provide further insight into the force-time curve differences between participants with low and high RPA$_{\%dec}$ scores respectively.

There is a lack of research on the underpinning characteristics of RPA and what determines low or high RPA$_{\%dec}$ scores. In research that investigated a range of potential predictors of RPA, a substantial amount of variance was left unexplained (*Natera et al., 2024*). Only 48.4% of the variance could be explained by field assessments of lower limb strength, maximal force and peak power output along with aerobic and repeated sprint abilities (*Natera et al., 2024*). With much still unknown about RPA, investigating the force-time curve differences between participants with low and high RPA$_{\%dec}$ might provide further information to practitioners and researchers on what phases of the jump are affected and what phases of the jump might differentiate between participants with low and high RPA$_{\%dec}$. With this understanding practitioners and coaches can monitor changes in RPA more effectively and may be able to design more targeted training programs to improve RPA performance.

The aims of this study were to identify changes in the force-time curve across distinct phases during performance of the LCMJ20. A secondary aim was to compare the force-time curve between participants with low RPA$_{\%dec}$ (effective ability to maintain peak power output) and high RPA$_{\%dec}$ (less effective ability to maintain peak power output). We hypothesised that (1) decreases in the braking and propulsive phase of the jump would occur across all three of the jump series (jump series 1 = jumps 2–5; jumps series 2 = jumps 9–12 and jump series 3 = jumps 16–19) within the LCMJ20 and (2) that differences in both the braking and propulsive phases of the jump would be found between participants with low and high RPA$_{\%dec}$, with low RPA$_{\%dec}$ participants showing less of a decrease in both the braking and propulsive phases of the jumps.

## MATERIALS & METHODS

### Participants

Eleven male, well-trained field hockey players (age $21.6 \pm 2.4$ years, body mass $78.2 \pm 6.8$ kg, stature $182.1 \pm 5.3$ cm) competing at state level, volunteered to participate in this study. A similar participant number has been used in several studies examining countermovement jump performance using 1-dimensional analysis methods (*Gathercole et al., 2015*; *Rice et al., 2017*; *Wu et al., 2019*). Ethical approval was granted by the Bond University Human Research Ethics Committee (N00156) and written consent was provided by all participants. Each player had a minimum of three years of resistance training experience, with heavy loaded squats and loaded jumps performed consistently throughout training. The investigation was performed in pre-season with each player undergoing the same training schedule and programme for the previous six weeks. The assessment procedures were explained in detail to all participants and a resistance training questionnaire was used to verify study eligibility. Participants who were currently injured, did not have the appropriate resistance training experience, or had not performed heavy squats and loaded jumps consistently for the 4 weeks prior to data collection, were excluded from the study.

### Study design

This study applied a single cohort observational single measure design. Differences in the force-time curve across three distinct series of jumps throughout the LCMJ20 were investigated. Jump clusters were selected to represent a range of jumps at the start of the assessment, where power output has been shown to be maintained (*Baker & Newton, 2007*), and then evenly positioned to capture jumps in the middle and at the end of the LCMJ20 (series 1 = jumps 2–5, series 2 = jumps 9–12, series 3 = jumps 16–19). As a secondary analysis, to be used to assist in the practical application of our force-time analysis approach, this study also investigated the differences in the force-time curve between participants in the 1st and 4th quartile, with these groups being described as having low RPA$_{\%dec}$ and high RPA$_{\%dec}$ scores, respectively.

### Procedures

Participants attended the strength and conditioning facility on two occasions. On the first occasion, descriptive data was collected and the three repetition maximum half squat (3RM

HS) was performed. Familiarization with the LCMJ20 assessment was then conducted and one week later the participants attended the second testing session where the LCMJ20 was performed.

A general warm up was performed at the start of each testing session prior to a specific warm up for each assessment. The general warm up consisted of 5 min of stationary cycling at a perceived exertion between 3–4 out of 10 on the Borg CRP-10 scale (*Zamunér et al., 2011*), followed by a range of dynamic flexibility exercises for the lower limb and trunk.

## Estimated one repetition maximum half squat

A 3RM HS assessment was conducted on a Smith machine with a full set of weight plates. The bar was placed on the upper trapezius muscles in a high bar placement, and the squat descent was monitored to a knee angle of 90° with the use of a goniometer. A thin rubber band was positioned so that when the required depth was reached the posterior thighs of each participant would contact the band (*Thomasson & Comfort, 2012*). As further precaution to monitor squat depth, a video recording from the sagittal plane was also used. The use of a Smith machine and the monitoring of squat depth were essential to match requirements for the RPA assessment (*Natera et al., 2023*).

After a specific warm up consisting of four sub-maximal sets of one repetition, each participant began gradually building to an estimated 3RM load (*Cronin & Hansen, 2005*). The participants then attempted their estimated 3RM and if they were successful, after three minutes of passive recovery, 5 kg was added until their 3RM was achieved (*Cronin & Hansen, 2005*). The last successful 3RM HS lifted was converted into an estimated 1RM.

In following the procedures of *Natera et al. (2023)*, the estimated 1RM was calculated using the average of seven different 1RM estimation formula's (*Brzycki, 1993*; *Epley, 1985*; *Lander, 1985*; *Lombardi, 1989*; *Mayhew et al., 1992*; *O'Connor, Simmons & O'Shea, 1989*; *Wathen, 1994*). It is considered safer to use an average of these different 1RM estimations to avoid validity issues influenced by the type of exercise performed and the population used (*Natera et al., 2023*). The mean estimated 1RM derived from these equations was used to quantify loading for the RPA assessment.

## Repeat power ability assessment

The RPA assessment consisted of 20 maximal CMJs with an estimated 30% 1RM load extrapolated from the 3RM HS assessment (*Natera et al., 2023*). The protocol used, including the number of jumps and the load, were consistent with the original research that found the LCMJ20 to be a reliable assessment of RPA (*Natera et al., 2023*). A set of dual force plates (ForceDecks; VALD Performance Sytems, Brisbane, Australia) sampling at 1,000 Hz was used to collect all RPA force-time data. After the general warm up, a specific warm up consisting of 3 LCMJs with the 30% 1RM load was performed five minutes prior to commencing the LCMJ20 assessment. The LCMJs during the specific warm up were performed at a perceived intensity of 50%, 75% and 100% and were each separated by 10 s. A "power standard" for each participant was established by extrapolating peak power output for the final warm up LCMJ with a linear position transducer (Gymaware, Lisborne, Australia).

A metronome was used to maintain rhythm for each jump of the LCM20 at 20 beats per minute, with the start of each jump commencing every 3 s. The timing of each jump allowed for precise data collection points whilst providing limited inter-repetition rest (∼1–1.5 s). A Smith machine was used with the barbell in a high bar squat position and the depth of each jump was visually and kinesthetically monitored to 90° knee flexion with the use of a thin elastic band. After landing from each jump, the participants immediately returned to a standing posture, with knees, hips and back extended, and awaited the sound of the metronome to initiate the next jump.

Force-time data was collected on the force plates whilst instantaneous feedback of derived peak power output was provided for each jump of the LCMJ20 using a linear position transducer. The instantaneous feedback was visually displayed on a tablet screen placed directly in front of the participant, whilst a live verbal reading of peak power output for each jump was also provided by the researchers. Data from the linear position transducer was used only for motivational purposes and was not used in any further analysis. The researchers also provided verbal motivation to the participants to perform every jump at maximal intent. It was essential for each participant's "power standard" to be attained within the first three jumps for the assessment to continue. Attaining the "power standard" within this time frame was essential to minimise the risk of participants employing a pacing strategy throughout the LCMJ20 (*Natera et al., 2023*) Given the high variability found in the first and last jumps in the LCMJ20, these two repetitions were removed from the final analysis (*Natera et al., 2023*).

## Data analysis

Two different procedures were performed to analyse RPA force-time data in this study. A traditional RPA approach, established by *Natera et al. (2023)* and a novel RPA data analysis approach used for the first time in the current study. In the traditional RPA data analysis approach, peak power output for jump 2 to 19 were derived from force-time data using customised MATLAB code (The MathWorks, Natick, MA, USA). Acceleration data was filtered using a second-order Butterworth filter, with a cutoff frequency of 2.5 Hz, to reduce drift caused by integrating derived acceleration data. An RPA peak power percent decrement score ($RPA_{\%dec}$) was then established using the following equation: $RPA_{\%dec} = 100 \times$ (total jump peak power output/ideal jump peak power) $- 100$. A full description of the traditional data analysis procedures can be found in *Natera et al. (2023)*.

In the novel approach, raw force plate data was again passed into a custom MATLAB script. The MATLAB script was written to extract and analyse each jump of a participant's 20 jump trial individually in order for the data to be further analysed using SPM. Only jumps 2–19 were used for further analysis, with all jumps normalised to a common trial length by adding data (force measured during standard quiet) prior to movement onset. The process for identifying the point of movement onset were based on procedures by *McMahon et al. (2018)* and *Natera et al. (2023)* (Fig. 1).

The data was normalised with respects to specific time points. Every jump was cut between unweighting and toe off. All unweighting to eccentric force pre-flight were compared for length and the longest from any jump was found. Then all jumps with a

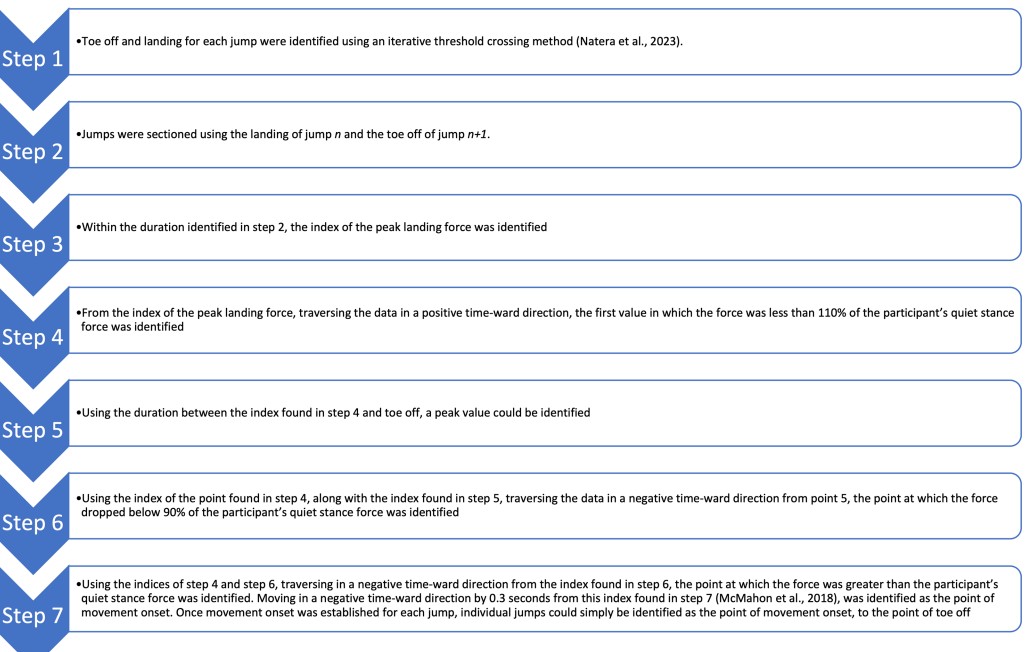

**Figure 1  Movement onset analysis process for the LCMJ20.** Each row describes a different step in the analysis process of the LCMJ20 force-time raw data to identify movement onset and toe off for each jump. LCMJ20, loaded countermovement jump assessment consisting of 20 repetitions.

shorter number of data points between the two points were interpolated to create a similar number of data points for every jump between unweighting and eccentric force pre-flight. This procedure was done for all sections, meaning that all key landmarks were aligned across the data sets. Values were interpolated with the spline function from stats package using the "fmm" method which is that of Forsythe, Malcolm and Moler (*Grund, 1979*). Documentation says an exact cubic is fitted through the four points at each end of the data, and this is used to determine the end conditions.

## Statistical analysis

The SPM process uses a series of linear models to compare the entire time series, with additional observations handled in a similar manner as a more conventional repeated measures ANOVA. An initial SPM one-way ANOVA was completed on the normalised time series to identify differences in the force-time curve across jump series 1, 2 and 3. Where significant effects ($a = 0.05$) were reported, *post hoc* analysis using an SPM paired samples $t$-test was used to compare between jump series.

As a secondary analysis used to provide practical relevance and further context to the force-time curve analysis results. The traditional RPA method was performed and participants were then ranked in quartiles based on their RPA$_{\%dec}$ scores. Quartiles were used to process the results of the traditional RPA method in order to identify the participants who performed at extreme ends of the LCMJ20 within this cohort. SPM analysis using a planned independent t-Tests was then conducted to examine differences across the

LCMJ20 between the 3 participants in 1st quartile and the 3 participants in the 4th quartile. Appropriate Bonferroni $p$ value adjusts were made for multiple comparisons.

## RESULTS

The SPM one-way ANOVA showed a significant difference ($p < 0.001$, $F* = 10.762$) across the three distinct jump series, during the propulsive phase (Fig. 2). This difference occurred between 74–98% of the time normalized force-time curve. An SPM paired samples $t$-test further highlighted significant differences between jump series 1 and 2 ($p < 0.001$, t* $= 3.995$) (Fig. 2, left column) and jump series 1 and 3 ($p < 0.001$, $t* = 3.750$) (Fig. 3, middle column). Significant differences between jump series 1 and 2 and jump series 1 and 3 occurred during the propulsive phase between 70–98% and 86–99% of the jump respectively. No significant difference was found between jump series 2 and 3 (Fig. 3, right column).

Using the traditional RPA analysis method (*Natera et al., 2023*), $RPA_{\%dec}$ scores ranging from 20.64 to 34.99% were found for the 11 participants. Quartiles were then calculated; 1st quartile $\leq$ 24.41%, 2nd quartile $>$ 24.41 and $\leq$ 26.57%, 3rd quartile $>$ 26.57 and $\leq$ 27.93% and 4th quartile $>$ 27.93%. SPM planned independent $t$-test found significant differences between the 1st quartile and the 4th quartile during the braking phase at 44–48% of the jump and during the propulsive phase at 74–94% of the jump ($p = 0.001$, $t* = 3.958$) (Fig. 4).

## DISCUSSION

The aims of this study were to understand whether greater insights into RPA performance could be achieved by replacing and comparing the discrete variable RPA analysis using $RPA_{\%dec}$, with a novel RPA analysis of the LCMJ20 force-time curve using SPM. In gaining a greater insight into RPA, through force-time curve analysis, practitioners wishing to enhance and develop RPA may gain a greater depth of understanding as to where changes in the force-time curve occur during the LCMJ20 and potentially where best to target training to enhance RPA.

We report that as the LCMJ20 assessment progresses there is a significant decrease in force output during the late propulsive phase (Fig. 3, left and middle column). This study also reports that participants with a low $RPA_{\%dec}$ (better RPA) displayed both higher braking (between 44–48% of the force-time curve) and propulsive forces (between 74–94% of the force-time curve) throughout the LCMJ20 than those with a high $RPA_{\%dec}$ (Fig. 4). Overall, SPM analysis provides further insights into where RPA decrements occur within the jump phases and also to differentiate between low and high $RPA_{\%dec}$ participants.

Between jump series 1 and jump series 2 there was a significant reduction in propulsive force during the LCMJ20 between 74–98% of the force-time curve (Fig. 2, middle column). The change in propulsive force between jump series 1 and 2 was consistent with fatigue induced deficits in power output observed during a set of 10 LCMJs (*Baker & Newton, 2007*). *Baker & Newton (2007)* reported a maintenance of power output for up to five jumps before there was a significant reduction in power output for the remaining jumps. Power

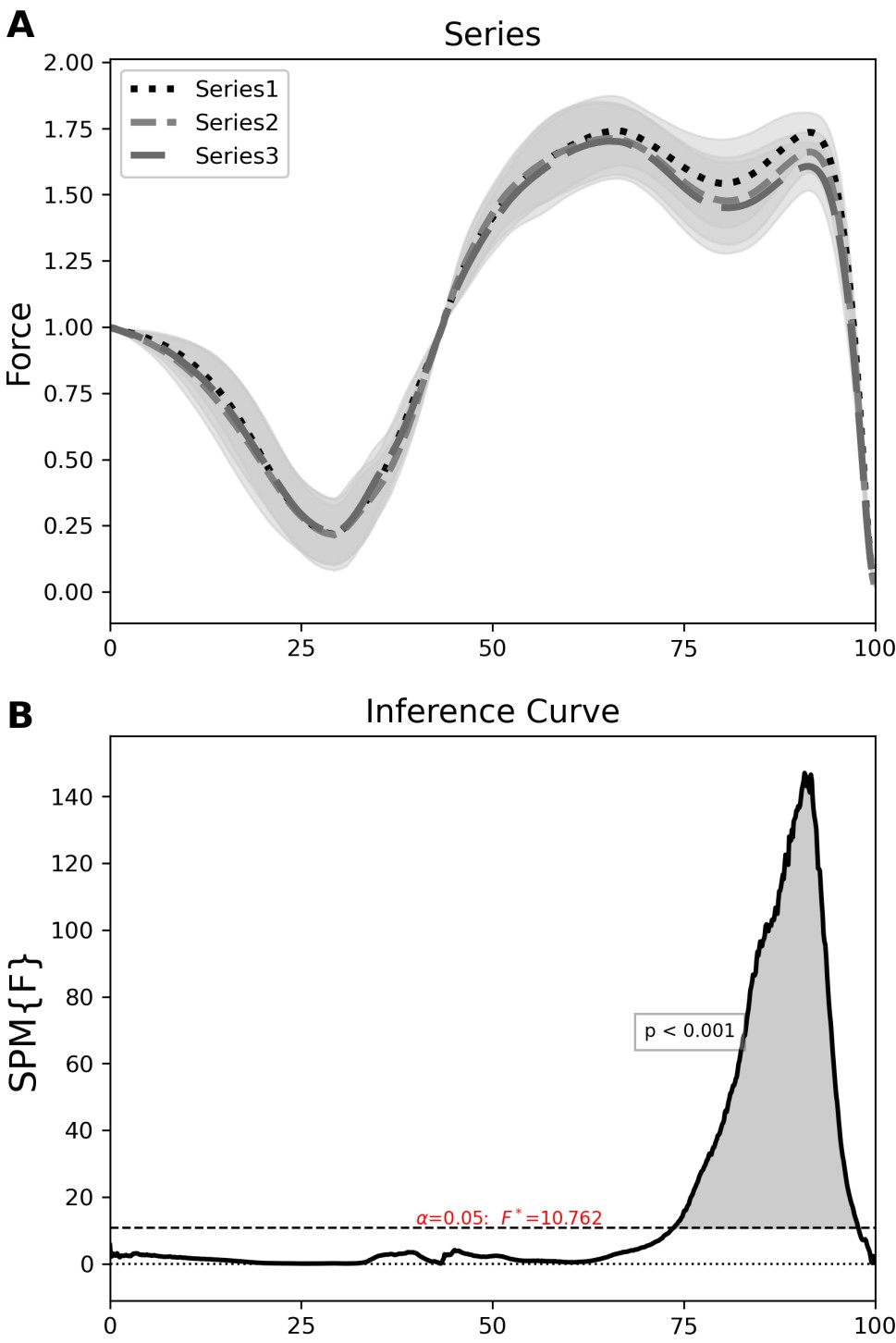

**Figure 2** **SPM one-way ANOVA results for the 3 jump series across the LCMJ20.** Top—normalised force time curves with SD cloud (shaded) for each jump series; jump series 1 (dotted black line), jumps series 2 (dashed grey line) and jump series 3 (full grey line). Bottom—inference curve with critical threshold (dashed line) and suprathreshold cluster (shaded) indicating a significant difference ($p < 0.001$). SPM, statistical parametric mapping; ANOVA, analysis of variance; LCMJ20, loaded countermovement jump assessment consisting of 20 repetitions; SD, standard deviation.

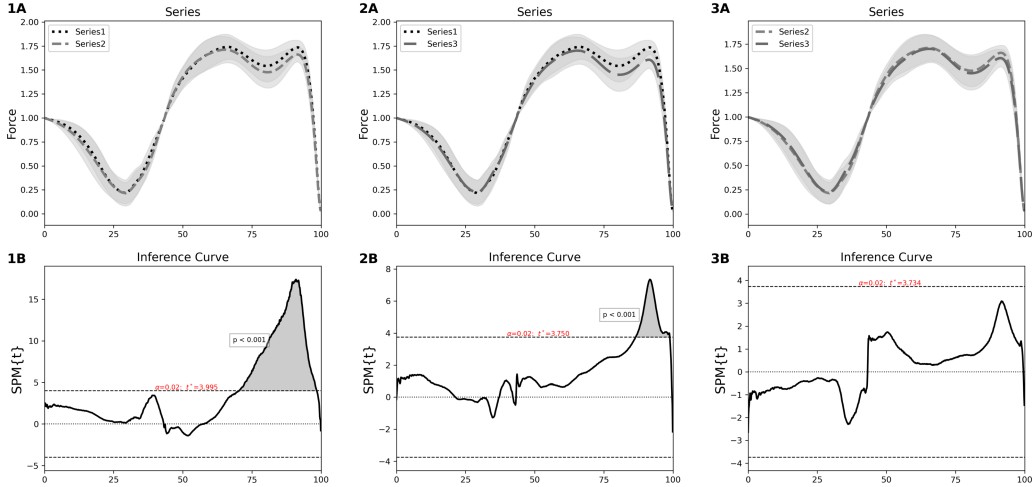

**Figure 3  SPM paired samples *t*-tests between jump series 1 and jump series 2 (1A and 1B), jump series 1 and jump series 3 (2A and 2B) and jump series 2 and jump series 3 (3A and 3B).** 1A, normalised force time curves with SD cloud (shaded) for jump series 1 (dotted black line) and jump series 2 (dashed grey line). 1B, inference curve with critical thresholds (dashed lines) and suprathreshold cluster (shaded) showing a significant difference ($p < 0.001$). 2A, normalised force time curves with SD cloud (shaded) for jump series 1 (dotted black line) and jump series 3 (full grey line). 2B, inference curve with critical thresholds (dashed lines) and suprathreshold cluster (shaded) showing a significant difference ($p < 0.001$). 3A, normalised force time curves with SD cloud (shaded) for jump series 2 (dashed grey line) and jump series 3 (full grey line). 3B, inference curve with critical thresholds (dashed lines), no suprathreshold clusters and no significant difference ($p > 0.001$). SPM, statistical parametric mapping; SD, standard deviation.

output, as measured during LCMJs, is a compound variable derived from the product of force and velocity during the propulsive phase of the jump (*Linthorne, 2021*). A reduction in propulsive force will tend to have a concomitant reduction in power output. With jump series 1 encompassing jumps 2–5 and jump series 2 encompassing jumps 9–12; a similar significant reduction in jump performance to that reported by *Baker & Newton (2007)* is observed in this study with propulsive force significantly different between jump series 1 to 2.

An interesting finding in this study is that propulsive force appears to remain relatively consistent with no further reduction beyond jump series 2, with no significant differences observed between jump series 2 and 3. Jump series 2 culminates after the 12th jump, which corresponds to 36 s of total time or ∼18 to 21 s of accumulated maximal effort time, with each individual jump in the LCMJ20 being ∼1.5–2 s (*Natera et al., 2023*). The full LCMJ20 is a 60 s assessment and encompasses a maximal effort time of ∼35 s. With maximal intensity exercise performed in these time frames it is possible that LCMJ20 results may be explained by metabolic processes. It is likely that phosphocreatine is the predominant energy source during jump series one, with only ∼7.5 s of maximal effort performed by the completion of series 1 (*Hargreaves & Spriet, 2020*). During jump series 2, maximal effort time has increased up to ∼21 s, resulting in a likely larger contribution of energy supply from anaerobic glycolysis (*Hargreaves & Spriet, 2020*). Whilst aerobic energy contribution will continue to increase to the end of the LCMJ20, anaerobic glycolysis is likely still the

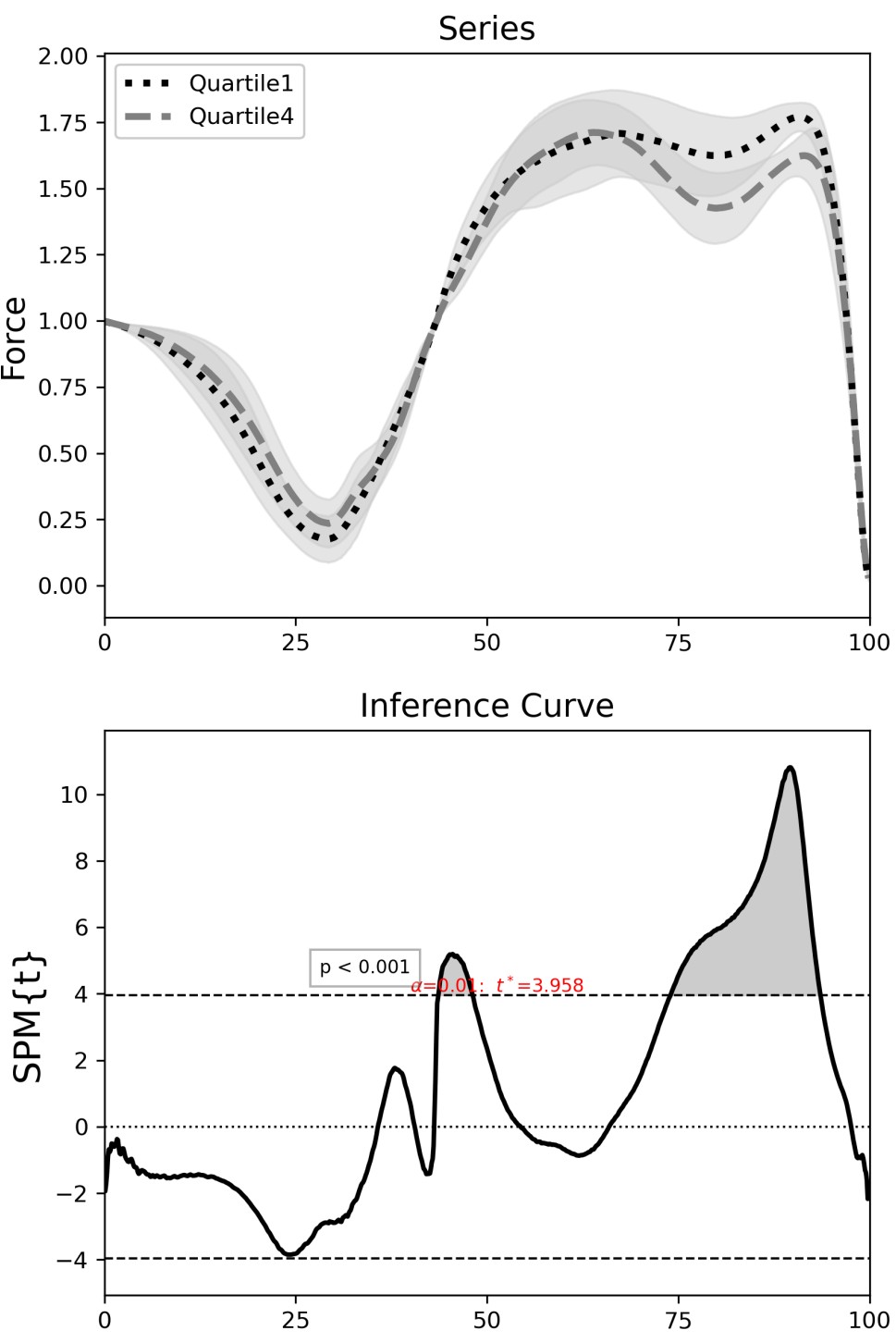

**Figure 4  SPM planned independent *t*-test between low RPA$_{\%dec}$ and high RPA$_{\%dec}$ participants across the LCMJ20.** Top—normalised force time curves across the LCMJ20 with SD cloud (shaded) for participants in Quartile 1 (low RPA$_{\%dec}$) (dotted black line) and participants Quartile 4 (high RPA$_{\%dec}$) (dashed grey line). Bottom—inference curve with critical thresholds (dashed lines) and suprathreshold clusters (shaded) showing significant differences ($p < 0.001$). SPM, statistical parametric mapping; RPA$_{\%dec}$, percent decrement of peak power output for the LCMJ20; LCMJ20, loaded countermovement jump assessment consisting of 20 repetitions; SD, standard deviation.

dominant energy system during jump series 3 (*Parolin et al., 1999*). The most significant change in energy system contribution is likely to occur between jump series 1 and jump series 2, where energy system predominance shifts from phosphocreatine hydrolysis to anaerobic glycolysis (*Hargreaves & Spriet, 2020*).

With strong relationships found between RPA and RHIEs (*Natera et al., 2024*), training to improve the maintenance of propulsive force between jump series 1 and jump series 2 is likely to be an important factor in RHIE performance. A number of RPA assessments have shown strong relationships with 30 s Wingate cycle performance; therefore, the anaerobic glycolytic system is likely to be the predominant energy system for RPA performance (*Bosco, Luhtanen & Komi, 1983*; *Fry et al., 2014*; *Sands et al., 2004*). Considering Yoyo intermittent recovery test 2 performance was not related to RPA performance (*Natera et al., 2024*), improving aerobic abilities may not enhance RPA performance. Improving anaerobic capacity through high intensity interval training may likely be one method of training to enhance RPA performance (*Atakan et al., 2021*; *Foster et al., 2015*; *Ko, Choi & Lee, 2021*; *Stöggl & Björklund, 2017*).

However, in research attempting to establish validity for the LCMJ20, the Bosco 30 s body weight jump test was found to have no association and poor levels of agreement (bias = 19.164, limits of agreement = 10.56–27.768) with LCMJ20 performance (*Natera et al., 2023*). It is suggested that the use of external load in the LCMJ20 may distinguish the rate and magnitude of power decline between the LCMJ20 and the Bosco 30 s jump test (*Natera et al., 2023*). Training methods to enhance LCMJ20 performance and the maintenance of propulsive force are likely to require the combined use of externally loaded ballistic exercises with set and repetition configurations that aim to predominantly stress anerobic glycolysis. One such method of training is high volume power training (HVPT), where 3–5 sets of 10–20 repetitions of ballistic exercise is performed with the intent to maintain power output for each repetition (*Natera, Cardinale & Keogh, 2020*). HVPT has previously been shown to enhance RPA, RHIE performance and anaerobic capacity and may be an effective training method to maintain propulsive force output across the LCMJ20 (*Apanukul, Suwannathada & Intiraporn, 2015*; *Bosco et al., 1994*; *Gonzalo-Skok et al., 2016*; *Romero-Arenas et al., 2018*).

When investigating differences across the LCMJ20 between participants with low $RPA_{\%dec}$ and high $RPA_{\%dec}$, significant differences are also found in the propulsive phase (Fig. 3). Participants with low $RPA_{\%dec}$ have better RPA and can therefore better maintain peak power output closer to maximal levels. Low $RPA_{\%dec}$ participants were found to have significantly higher propulsive forces across the LCMJ20. This finding is despite CMJ maximal peak power, isometric midthigh pull maximal force output and 3RM HS strength all showing no association with RPA (*Natera et al., 2024*). Therefore, participants with low $RPA_{\%dec}$, who exhibit significantly greater propulsive force across the LCMJ20, do not simply have greater maximal or rapid force capabilities. The greater propulsive forces found in the low $RPA_{\%dec}$ participants is due to their ability to maintain propulsive forces rather than their maximal or rapid force generating capability.

A contributing factor to low $RPA_{\%dec}$ participants better maintaining propulsive force, may be their ability to utilise the braking phase more effectively in allowing lower limb

muscles to build a higher active state in order to enhance propulsive forces (*Bobbert et al., 1996*; *Cormie, McGuigan & Newton, 2010*). In the current study, low RPA$_{\%dec}$ participants showed significantly higher braking forces in comparison to high RPA$_{\%dec}$ participants. Whilst low RPA$_{\%dec}$ participants were able to produce higher braking forces throughout the LCMJ20, high RPA$_{\%dec}$ participants were less efficient at utilising the braking phase of the jump and may also have avoided high braking forces in an attempt to conserve energy for the propulsive phase of the jump (*Buchheit et al., 2010*; *Jidovtseff et al., 2014*).

In an investigation of different jump strategies *Jidovtseff et al. (2014)* found a reduction in velocity and acceleration during the braking phase of LCMJs in comparison to CMJs without load. It is suggested that this strategy may be employed to reduce loading and to minimise the risk of injury (*Jidovtseff et al., 2014*), with this potentially even more important with high levels of neuromuscular fatigue. Although this strategy may have been used to minimize risk of injury, this strategy might also be progressively employed by high RPA$_{\%dec}$ participants as a strategy to minimise fatigue as the LCMJ20 progressed. Low RPA$_{\%dec}$ participants may have an enhanced ability to tolerate high velocity and acceleration during the braking phase of the LCMJ20 and were therefore able to maximise the stretch shortening cycle to optimise propulsive forces across the LCMJ20. This is an important finding, as a multiple linear regression including a RHIE assessment consisting of maximal shuttle runs with multiple sprint accelerations, decelerations and 180° change of directions has been found to be the best predictor of RPA (*Natera et al., 2024*). Not only is the shuttle run assessment an assessment of repeated maximal efforts, but it is also an assessment where repeated maximal eccentric loading is experienced in the decelerations and change of directions used during the assessment. The decelerations and change of directions are known to come at a high physiological cost and it is a similar physiological cost that may differentiating between participants that have the capacity to repeatedly produce high eccentric braking forces in the LCMJ20 (*Buchheit et al., 2010*; *Dos'Santos et al., 2017*; *Lake et al., 2021*).

Several limitations need to be acknowledged in this study. Despite there being a number of similar studies with similar participation numbers using 1-dimensional analysis methods to investigate countermovement jump performance (*Gathercole et al., 2015*; *Rice et al., 2017*; *Wu et al., 2019*), participant numbers may still be considered relatively low and this should be considered in the interpretation of the results. Our secondary analysis of the differences between the 1st and 4th quartile, low and high RPA$_{\%dec}$, means an even smaller sample size was used in this analysis. With the small sample size used in this research, it must be acknowledged that the procedures used in this study may not produce the same results in replicated studies using larger cohorts. Furthermore, in the comparison between the low and high RPA%dec groups, it must be acknowledged that the small window of difference in the braking phase, observed between 44–48%, may be partly attributed to the Smith machine bar flexing and rebounding as a result of the vertical change in direction. Whilst this research focused primarily on the force-time contributions to power output across the unweighting, braking and propulsive phases of the jump, it is important to acknowledge that the landing phase of the LCMJ20 may have also contributed to RPA performance. It is likely that repeated landings with external loads and the utilisation of different landing
strategies may influence fatigue levels and concomitantly affect jump strategy (*Lake et al., 2021*). Despite controlling for jump timing and jump depth the lack of analysis on jump phase duration may be another limitation to this study. Changes in jump phase duration may have contributed to changes in performance across the 3 jump series and differences between participants with high and low $RPA_{dec\%}$. Future research should examine the effect of the landing phase and changes in phase duration and also seek to identify energy system contribution across the LCMJ20. Research should also look to enhance RPA by examining training interventions like HVPT, that may affect the maintenance of both braking and propulsion phase force across the LCMJ20. Likewise, future research can also look to identify smallest worthwhile change values to better inform practitioners on the effectiveness. of such training interventions.

Force-time curve analysis on the LCMJ20 using SPM appears to provide greater insights into RPA performance than traditional zero-dimensional analysis approaches. In particular, the reduction in propulsive force between jump series 1 and 2 provides evidence as to when fatigue occurs and what phase of the jump is most affected. An understanding of the differences between low and high $RPA_{\%dec}$ participants presents some preliminary insight into the force-time contributions needed to maintain RPA more effectively across the LCMJ20.

Whilst SPM analysis provides greater insights for the practitioner, traditional RPA analysis is a useful and informative quantitative measure of RPA. Due to there being significant differences in the force-time curve between low and high $RPA_{\%dec}$ participants, it is evident that traditional RPA analysis provides quantifiable information that can differentiate participants ability to maintain peak power output across the LCMJ20. It is recommended that both the traditional analysis of RPA and the more novel force-time curve analysis be used in conjunction, to provide the practitioner with both quantitative information and a more comprehensive description of RPA. A more comprehensive description of RPA will inform the practitioner as to when during the LCMJ20 and during what phases of the jump changes occur. This additional detail may likely lead to more effective training interventions to improve RPA performance.

# CONCLUSION

In attempting to gain greater insights into RPA by using force-time curve analysis, we found significant differences in the propulsive phase of the LCMJ20. Significant reductions in propulsive force were observed between jump series 1 (jumps 2–5) and jumps series 2 (jumps 9–12), within the LCMJ20. Participants who were able to maintain peak power output closest to their maximal values produced significantly greater braking and propulsive forces than participants who had high decrements in peak power output across the LCMJ20.

Whilst force-time curve analysis offers insights into RPA, traditional analysis of RPA using a percent decrement, of extrapolated peak power output, is a useful analysis that differentiates participants with low and high RPA and offers a quantitative measure of RPA performance that is reliable and related to RHIE performance. Practitioners can use the traditional RPA analysis method as the primary analysis of RPA and then use force-time

curve analysis as a secondary analysis to inform training and monitor changes in the force-time curve as a result of training interventions.

Based on the force-time curve analysis used in this study, where reductions in propulsive force were observed between jump series and reductions in braking and propulsive force were observed between high and low RPA$_{\%dec}$ participants, a training intervention that targets the maintenance of braking and propulsive forces is likely to improve RPA performance. With both aerobic and repeated sprint performance showing no associations with RPA (*Natera et al., 2024*), HVPT may be an effective training intervention to improve the maintenance of braking and propulsive forces across the LCMJ20 and enhance RPA. The duration and volume of maximal intensity efforts used in HVPT protocols may provide a specific stimulus to improve the maintenance of both braking and propulsive forces. Improving the maintenance of braking and propulsive forces is likely to shift a participant with a high RPA$_{\%dec}$ to a lower RPA$_{\%dec}$ with a resultant improvement in RPA.

## ACKNOWLEDGEMENTS

We would like to acknowledge the contribution of Ben Hindle in writing the MATLAB code for jump analysis and data preparation for SPM analysis.

### Funding
The authors received no funding for this work.

### Competing Interests
Justin W.L. Keogh is an Academic Editor for PeerJ.

### Author Contributions
- Alex O. Natera conceived and designed the experiments, performed the experiments, analyzed the data, prepared figures and/or tables, authored or reviewed drafts of the article, and approved the final draft.
- Steven Hughes analyzed the data, prepared figures and/or tables, authored or reviewed drafts of the article, and approved the final draft.
- Dale W. Chapman conceived and designed the experiments, authored or reviewed drafts of the article, and approved the final draft.
- Neil D. Chapman conceived and designed the experiments, authored or reviewed drafts of the article, and approved the final draft.
- Justin W.L. Keogh conceived and designed the experiments, authored or reviewed drafts of the article, and approved the final draft.

### Human Ethics
The following information was supplied relating to ethical approvals (*i.e.*, approving body and any reference numbers):

Bond University Human Research Ethics Committee.

## Data Availability

The data is available at figshare: Natera, Alex (2024). Normalised LCMJ20 data set formatted for SPM analysis. figshare. Dataset. https://doi.org/10.6084/m9.figshare.25133297.v1.

Natera, Alex (2024). SPM Code. figshare. Dataset. https://doi.org/10.6084/m9.figshare.25133228.v1.

Natera, Alex (2024). MATLAB Code for LCMJ20. figshare. Dataset. https://doi.org/10.6084/m9.figshare.25133126.v1.

Natera, Alex (2024). Raw LCMJ20 Force-Time Data. figshare. Dataset. https://doi.org/10.6084/m9.figshare.25133009.v1.

## Supplemental Information

Supplemental information for this article can be found online at http://dx.doi.org/10.7717/peerj.17971#supplemental-information.

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
