# Peer review of "Changes in the force-time curve during a repeat power ability assessment using loaded countermovement jumps"

_PeerJ, doi:10.7717/peerj.17971_

## Round 0.1 · original submission · Minor Revisions

The reviewers are generally positive about the manuscript and although there are a number of comments to address these are mainly points of clarification and detail to be added.

From my perspective there are a few points (also raised by reviewers) that I think it are relevant to consider/comment on in the manuscript.
1. Introduction - "One dimensional analysis, using Random Field Theory, is said to reduce bias and Type I error by avoiding the use of separate inferential tests at each time point" - I believe that this paper is best to support this statement as it specifically quantifies false positives (note for full disclosure I am a co-author on the paper).
Pataky, T. C., Vanrenterghem, J., & Robinson, M. A. (2016). The probability of false positives in zero-dimensional analyses of one-dimensional kinematic, force and EMG trajectories. Journal of biomechanics, 49(9), 1468-1476.
2. The analysis of quartiles - generally converting a continuous measure into categorical data is undesirable. In the statistics literature there are some papers that explore this and generally it seems as though a regression analysis is preferred where possible. For your analysis you could regress the RPA%dec measure over the CMJ profile to see if variation in RPA%dec is represented by differences in the CMJ profile (see https://spm1d.org/doc/Stats1D/onetwosample.html#regression)

See example papers below.
Altman, D. G., & Royston, P. (2006). The cost of dichotomising continuous variables. Bmj, 332(7549), 1080.
Maxwell, S. E., & Delaney, H. D. (1993). Bivariate median splits and spurious statistical significance. Psychological Bulletin, 113(1), 181–190. https://doi.org/10.1037/0033-2909.113.1.181
Royston, P., Altman, D. G., & Sauerbrei, W. (2006). Dichotomizing continuous predictors in multiple regression: a bad idea. Statistics in medicine, 25(1), 127-141.

3. Figures 3-6 are a bit repetitive. These could easily be combined into one 2 x 4 or 2 2x2 figures.
4. Small sample size should be recognised in the limitations
5. It is not clear in the methods exactly what data is taken to the SPM analysis. Are participant means calculated, then analysed with spm1d? Please provide more detail about the specific data extraction to analysis.

If you are willing to revise the manuscript please provide a point-by-point rebuttal to all comments raised.

Reviewer 1 ·

Basic reporting

Ln 71 – a supporting reference may be useful to help the reader e.g. Preatoni, E, Hamill, J, Harrison, AJ, Hayes, K, Van Emmerik, RE, Wilson, C, and Rodano, R. Movement variability and skills monitoring in sports. Sports Biomech 12: 69–92, 2013

Ln 77 if depth and timing are controlled during the LCMJ20, how would a change in jump strategy occur?

Ln 79 this point was made in the previous paragraph – can this written more concisely?

Ln 84 can more information be given here, it assumes the reader will easily understand the point

Ln 120 confusing on initial the read ‘….all three of the jump series.’ Could more detail be provided here?

Ln 144 with 11 participants, why use these quartiles and therefore exclude so much data?

Ln 201 can equipment details be provided?

Ln 242 how adequately powered was the study, with a sample of 11.

Ln 248 how was the shape of the force-time curve controlled for e.g. bimodal vs unimodal

Ln 257 how can 44-48% be explained? such a small window

Ln 267 what changes in jump strategy where observed in the study?

Ln 270 it would be helpful if the figure was cited and more detail was provided e.g. percentages

Ln 276 citing the figure again would help

Ln 288 would the power of the study be a limitation here?

Ln 303 can any physiological data be cited related to the specific test?

Ln 331 citing the figure again would help

Ln 351 the paper cited alluded to reduced injury but the sentence in this paper attempts to convey a different message

Ln 359 strange to start a paragraph with this point – it is disconnected from the point

Ln 372 what would be a suitable number to identify small worthwhile changes?

Ln 404 how has this paper controlled for those with higher power outputs simply having larger % decreases? especially when trying to increase performance

Experimental design

The arbitrary use of quartiles to process the data hasn't been justified with a sound rationale

Validity of the findings

The small sample size raises some issues regarding the validity of the findings.

Additional comments

no comment

·

Basic reporting

Overall, a few very minor things that it would be beneficial to provide some clarity or expansion upon. Please see below.

In lines 62-65, are there any available norms or recommendations on RPA decrement scores? What constitutes a high performing and poor performing score, respectively?

In lines 67-73, I understand and agree with the sentiment of this paragraph. However, I wouldn’t say that mean force is derived from an instantaneous point on the force-time curve. Mean force is generally depicted as the average force over a window of time with a defined starting and ending point. Wouldn’t analyzing mean force be an examination of the whole, or at least a large portion, of the force-time curve, like you are saying in lines 71-73?

In lines 77-79, as depth and jump timing are controlled, what variables could contribute to a change in jump movement strategy that could influence things like peak power output? Would this be something like changes in phase duration? While not the same as the assessment in the present study, previous work has demonstrated that the presence of fatigue has altered CMJ eccentric and concentric phase duration (DOI: https://doi.org/10.1123/ijspp.2013-0413), and phase duration without a change in CMJ depth (DOI: 10.23736/S0022-4707.17.06848-7). I think a bit more of explanation of this would be beneficial to the introduction.

In lines 119-122, it would be beneficial to have a directional hypothesis. Did you expect to see increases or decreases in these variables across the jump series and between high and low RPA performers?

In the description for figure 2 on page 29, where it says “ANNOVA = analysis of variance,” should ANNOVA be ANOVA?

Experimental design

Experimental design and methodology were very good. Just a few small questions I have that can very easily be addressed. Please see below.

Why 30% 1RM for the RPA assessment? Is this the load in which peak power is achieved during the jump squat exercise? Peak power during the jump squat exercise has been demonstrated to occur at a variety of loads, depending on how it was measured. McBride et al. found that peak power during the jump squat occurred at 80% of squat 1RM if measuring barbell peak power, but occurred at 0% of squat 1RM is measuring system peak power on a force plate (DOI: 10.1080/02640414.2011.587444). Other methods, looking at percentage of jump squat 1RM found that jump squat peak power occurred at 20% jump squat 1RM (DOI: 10.1519/JSC.0b013e318234ebe5). An explanation as to why 30% 1RM loads were used would be beneficial.

Line 206, why was it essential to reach the power standard within the first three jumps? Was this to ensure participants were giving maximum effort throughout the duration of the test?

Starting at line 244, did the distinct phases of the jump curves line up when they were normalized? For example, transitioning from braking to propulsive, did they occur at the same relative time points on the normalized curves? Did jump strategies (braking and propulsive phase duration) change over the course of the jump series?

Lines 253-258, Do you have the actual ranges of scores within the interquartile ranges? Is n of 11 enough to split into quartiles? How different is someone at the bottom of QT 1 and the top of QT 2 for example? Would it be possible to provide the actual RPA decrement scores for the participants in each range as part of the supplemental data?

Validity of the findings

In assessing the discussion and conclusion sections, I believe they were well written and informative. I have a few simple questions that can be easily addressed.

Lines 270-272 Where are you identifying the initiation of the propulsive phase? In normalizing the curves, do the windows for the braking and propulsive phases occur at the at the same relative percentage between the different jump series?

Lines 330-334 In assessing factors that contribute to maintenance of RPA, did you analyze durations of the braking and propulsive phases? Were there differences in braking phase duration and propulsive duration between series 1, 2, and 3, or between the high and low RPA performers? Previous work has shown that phase duration can change in the presence of fatigue, which may influence performance and may help to explain some of the differences seen. Normalizing all of the force-time curves to relative percentages of jump completion may negate some of the differences in performance that may be seen during the braking and propulsive phases across jump series or between high and low RPA decrement scores.

Line 382 Did the initiation of the propulsive phase occur at the same normalized time point across participants? I think it would be beneficial to discuss where on the normalized force-time curves the initiation of braking and propulsion began/ended and if that changed at all with fatigue. This would also indicate whether or not the durations of the braking and propulsive phases altered throughout the test.

How specifically can the force-time curve analysis be used to inform training? While it was evident that reductions in propulsive force were present between series 1 and 2, and between 1 and 3, it seems that improving energy system development and anaerobic capacity are the primary means of improving RPA performance (lines 312-314, 320-328). Was there anything specific to the jump performance/strategy that contributed to the decrements in propulsive force throughout the test, and how could changes in jump strategy throughout the test be used to inform training?

Additional comments

I believe this was an extremely well written and informative paper on a new and novel assessment of repeated anaerobic power output. The authors do a very good job at relating how performance during the RPA and how changes in performance may be related to exercise/sport performance. I believe with very minor revision to the areas addressed above that this paper should be accepted for publication.

---

## Round 0.2 · Minor Revisions

Thank you for your revised manuscript. Please address two final points:
1. Reviewer 1 - please refer to the sample size in the limitations
2. Reviewer 2 - please update Natera et al., 2003 to 2023.

Reviewer 1 ·

Basic reporting

no comment

Experimental design

citing other studies that have used small sample sizes is not a justification to also do so - not a major issue, it just needs to recognised as a limitation.

Validity of the findings

no comment

Additional comments

all comments considered by the authors - an excellent paper overall.

·

Basic reporting

No comment

Experimental design

No comment

Validity of the findings

No comment

Additional comments

I believe the authors adequately and appropriately addressed the comments brought up by reviewers and I would recommend that the article be accepted for publication.

I did find one small potential typo on line 200 under the "Repeat Power Ability Assessment" paragraph. The reference that reads (Natera et al., 2003), should this be (Natera et al., 2023)?

---

## Round 0.3 · accepted · Accept

Thank you for addressing all reviewer (and editor) comments thoroughly - I am fully satisfied with all revisions made and I am recommending the manuscript for publication.